# Beyond Low-rank Decomposition: A Shortcut Approach for Efficient On-Device Learning

**Le-Trung Nguyen** [1]    **Aël Quélennec** [1]    **Van-Tam Nguyen** [1]    **Enzo Tartaglione** [1]

## Abstract

On-device learning has emerged as a promising direction for AI development, particularly because of its potential to reduce latency issues and mitigate privacy risks associated with device-server communication, while improving energy efficiency. Despite these advantages, significant memory and computational constraints still represent major challenges for its deployment. Drawing on previous studies on low-rank decomposition methods that address activation memory bottlenecks in backpropagation, we propose a novel shortcut approach as an alternative. Our analysis and experiments demonstrate that our method can reduce activation memory usage, even up to $120.09\times$ compared to vanilla training, while also reducing overall training FLOPs up to $1.86\times$ when evaluated on traditional benchmarks. The code is available at https://github.com/Le-TrungNguyen/ICML2025-ASI.git.

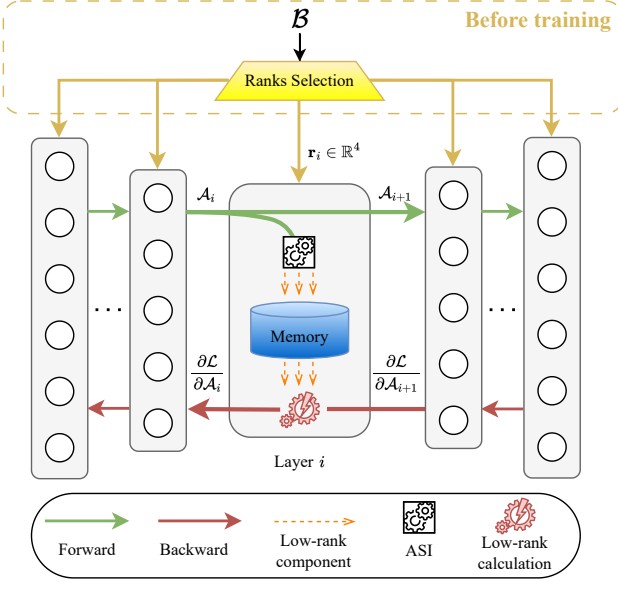

*Figure 1.* Trajectory of activation maps during a single training step of a deep learning model using ASI.

## 1. Introduction

Recent significant advances in deep learning have allowed artificial intelligence to penetrate deeply into human activities, enhancing the quality of life for users more than ever before. Deep learning has found applications in a wide variety of fields, such as healthcare through tumor image detection (Hu et al., 2018) or finance through quantitative trading (Ozbayoglu et al., 2020). Most notable are applications of large language models (LLMs), such as GPT (Radford et al., 2018) or Gemini (Gemini Team et al., 2023).

In that regard, one of the current prominent research branches is on-device learning, where models are fine-tuned directly on edge devices, eliminating the need for commu-

nication between data and servers, and thereby enhancing user data security. This method proves valuable in diverse scenarios, particularly in environments without internet connectivity, such as autonomous vehicles (Dhar et al., 2021). It can also enable smartphones to learn from local user data, allowing personalized adaptation to individual usage patterns (Hard et al., 2018).

Along with the substantial increase in the number of parameters in deep learning models comes an exponential rise in resource demand to train and exploit such models. During training, the main algorithmic bottleneck with respect to energy and memory is backpropagation. Although it has demonstrated its effectiveness in terms of performance, surpassing many other methods such as Forward-Forward (Hinton, 2022) and PEPITA (Pau & Aymone, 2023), it is not yet a "resource-friendly" algorithm. Specifically, for each trained layer, backpropagation requires storing in memory the corresponding activation maps during the forward pass (Gomez et al., 2017), which are then used to compute weight derivatives during the backward pass. As the number of model

---

[1]LTCI, Télécom Paris, Institut Polytechnique de Paris, France. Correspondence to: Le-Trung Nguyen <le-trung.nguyen@telecom-paris.fr>, Enzo Tartaglione <enzo.tartaglione@telecom-paris.fr>.

*Proceedings of the 42nd International Conference on Machine Learning*, Vancouver, Canada. PMLR 267, 2025. Copyright 2025 by the author(s).

parameters increases, the memory required to store these parameters and activations during training grows accordingly.

The increase in resource demands leads to longer training times and higher demands for data centers, resulting in greater carbon footprint emissions (Strubell et al., 2020; Thompson et al., 2020; Patterson et al., 2021). This critical issue demands immediate attention. The techniques used to train models directly on devices offer a promising solution, as their inherently low energy consumption helps address these challenges (Alajlan & Ibrahim, 2022).

Various research directions have emerged to overcome back-propagation inefficiencies and enable learning directly on devices. A pioneering study, initiated by Lin *et al.*, demonstrated that fine-tuning a predefined subnetwork of a deep model under a strict memory constraint of just 256KB while maintaining acceptable performance is entirely feasible (Lin et al., 2022). In a similar fashion, Quélennec et al. (2024) propose to dynamically fine-tune the most suitable subnetwork at each training step rather than relying on a fixed one, allowing for improved accuracy while maintaining memory usage under constraint. Extending beyond the scope of on-device learning, many other studies have been proposed to optimize the training process by reducing the number of trained parameters, notably LoRA (Hu et al., 2021) and its variants. While these approaches significantly reduce the number of parameters that need updating during training, they all neglect activation memory constraints. To solve this issue, Nguyen et al. (2024) applies low-rank strategies to compress activation maps for a greatly reduced memory footprint with a limited performance drop. Although promising, the actual on-device feasibility of their method is compromised by the prohibitive computational cost of performing the activation decomposition between each training epoch, as well as their lack of control over memory constraints.

Inspired by previous studies on the stability of activation maps during training (Virmaux & Scaman, 2018), we propose ASI (**A**ctivation **S**ubspace **I**teration) to address the issue of reducing computational complexity while training the model (Fig. 1). By identifying the most suitable decomposition rank and projector once before training, we can effectively gain both in terms of memory occupation and computation. Leveraging gradient computation in the compressed space, we are also able to reduce backpropagation computation, as showcased when deploying our solution on small devices like a RaspBerry Pi 5.

The key contributions of our work are below.

- We propose a rank selection strategy to determine the most suitable ranks for each fine-tuned layer under a given budget constraint before training begins (Sec. 3.3). This is achieved by an estimation of the activation perplexity that accounts for the impact of the low-rank decomposition during back-propagation.

- We introduce the use of a single subspace iteration as a replacement for traditional compression methods, aiming to balance accuracy and computational efficiency (Sec. 3.4), translating into a consistent reduction in computation time (Sec. 3.5).

- We demonstrate the effectiveness of our method through various experiments, including ImageNet-1k, and performing real measurements on a resource-limited device like a Raspberry Pi 5 (Sec. 4).

## 2. Related Works

In this section, we will first present some recent works that propose low-rank decomposition (Sec. 2.1). Although conceptualized for reducing the dimensionality of the parameters representation, these set methods and tools enable the design of strategies for activation map decomposition. Next, in Sec. 2.2, we discuss previous works that directly address our target problem and highlight their limitations.

### 2.1. Low-rank Decomposition for Weight Compression

Low-rank approximation methods aimed at reducing the number of parameters required for training have demonstrated significant effectiveness with approaches like LoRA (Hu et al., 2021). By training a low-rank adapter, the number of trainable parameters is reduced by up to $10,000\times$ without introducing additional inference latency.

Zhang et al. (2023) have shown that assigning the same rank to all fine-tuned layers is not an advisable strategy, as different layers contribute differently to the model's performance. To address this issue, they proposed AdaLoRA, which dynamically allocates suitable ranks to each fine-tuned layer at each training step while adhering to a predefined parameter budget. Subsequent studies, such as DyLoRA (Valipour et al., 2022), SaLoRA (Hu et al., 2023), SoRA (Ding et al., 2023), and ALoRA (Liu et al., 2024), have similarly attempted to solve this problem by adaptively distributing ranks based on the importance of each layer to the performance of the model. A practical and effective solution emerged with the introduction of ASVD (Yuan et al., 2023). In this work, they propose analyzing the sensitivity of each layer to different compression rates (*i.e.*, the memory ratio between the low-rank version and the original tensor). Before training, they performed a binary search to determine the most suitable rank for each layer that satisfies a predefined budget, inspecting the distribution of the generated activations and compressing the parameters to minimize the induced distributional shift on the produced output.

All of these methods are specifically designed for weights

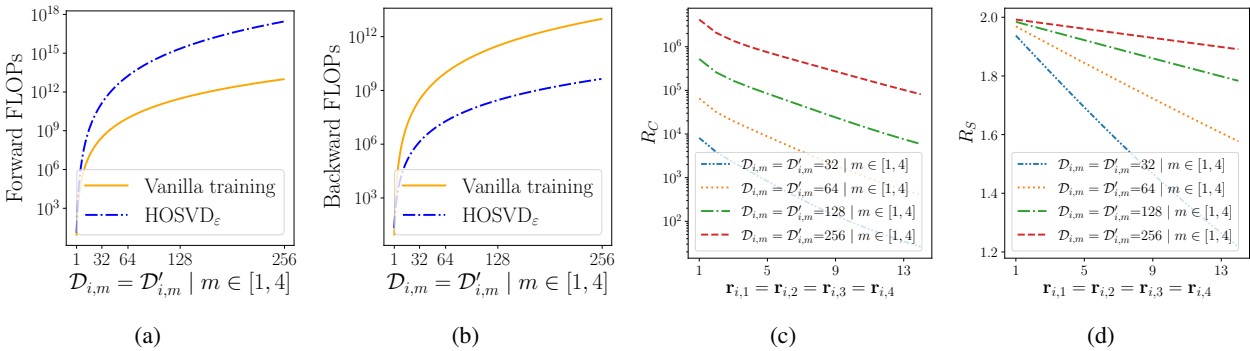

Figure 2. For convolutional layer $i$ with a single data batch of size $B$, **(a)** and **(b)** illustrate the predicted changes in the total number of FLOPs required to perform a forward pass and a backward pass, respectively, when comparing $\text{HOSVD}_\varepsilon$ with vanilla training. **(c)** and **(d)** show the predicted changes in compression rate $R_C$ and speedup ratios $R_S$ as functions of $\mathbf{r}_{i,m}$, when comparing ASI with vanilla training.

and cannot be applied directly to activation maps, partially solving the problem of memory and computation reduction when deploying the targeted deep model on-device.

## 2.2. Activation Map Compression

Recently, some works appeared in the context of activation map compression for convolutional neural networks, where activations pose a big issue. Recognizing the severe memory bottleneck in backpropagation, Yang et al. (2023) proposes gradient filtering as a possible solution. Given a predefined patch size, they applied pooling operators to it to generate approximated versions of activation maps and gradients. Their goal is to reduce resource consumption, specifically memory usage and FLOPs. However, relying solely on pooling techniques generally distorts the structured properties of tensors in deep neural networks. Moreover, gradient approximation propagates error during the backward pass, leading to a significant performance drop as the number of fine-tuned layers increases.

Later, Nguyen *et al.* proposed replacing pooling techniques with Higher-Order Singular Value Decomposition (HOSVD) based on an explained variance threshold $\varepsilon$ ($\text{HOSVD}_\varepsilon$), compressing only the activation maps (Nguyen et al., 2024). This method achieved impressive compression rates while preserving the structural properties of tensors and controlling information loss caused by compression. However, $\text{HOSVD}_\varepsilon$ incurs extremely high computational costs due to the HOSVD computation required at each training iteration, limiting its applicability due to high computation overheads. Besides, this approach lacks hard memory control (the explained variance is fixed, meaning that depending on the variance of the activations it can be more or less effective memory-wise).

Our proposed ASI goes beyond the limits posed by both Yang et al. (2023) and Nguyen et al. (2024): through the

design of an activation perplexity measure, we estimate the best ranks only once before training to ensure a strict constraint on memory consumption. Additionally, we leverage a single subspace iteration to effectively estimate the low-rank components of activation maps and perform low-rank gradient calculations, significantly reducing the overall training cost.

## 3. Method

In this section, after recalling backpropagation and its impact on memory consumption (Sec. 3.1) and low-rank decomposition through HOSVD (Sec. 3.2), we present how to perform rank selection in a compressed subspace for activation maps (Sec. 3.3) and how to apply it to effective computation and memory savings (Sec. 3.4), followed by a computational complexity analysis (Sec. 3.5).

### 3.1. The Memory Bottleneck in Backpropagation

Let us assume we work with a deep convolutional architecture.[1] We note with $i$ the index of a layer in a deep neural network, characterized by a $D_i \times D_i$ kernel $\mathcal{W}_i \in \mathbb{R}^{C_i' \times C_i \times D_i \times D_i}$. This layer processes an input tensor $\mathcal{A}_i \in \mathbb{R}^{B \times C_i \times H_i \times W_i}$ with $C_i$ channels and produces an output tensor $\mathcal{A}_{i+1} \in \mathbb{R}^{B \times C_i' \times H_i' \times W_i'}$ with $C_i'$ channels. Here, $H_i$ and $W_i$ represent the height and width of each input element, while $H_i'$ and $W_i'$ denote the corresponding dimensions of the output, and the variable $B$ indicates the batch size. For convenience, here on we write $\mathcal{D}_i = \{B, C_i, H_i, W_i\}$ and $\mathcal{D}_i' = \{B, C_i', H_i', W_i'\}$.

Let us consider the backward pass at this layer, where the

---

[1]This is one case of interest as activation maps in convolutional layers are typically very large. The same conclusions we reach can be straightforwardly obtained for fully-connected layers.

chain rule of backpropagation is computed as follows:

$$\frac{\partial \mathcal{L}}{\partial \mathcal{W}_i} = \frac{\partial \mathcal{L}}{\partial \mathcal{A}_{i+1}} \cdot \frac{\partial \mathcal{A}_{i+1}}{\partial \mathcal{W}_i} = \text{conv}\left(\mathcal{A}_i, \frac{\partial \mathcal{L}}{\partial \mathcal{A}_{i+1}}\right) \quad (1)$$

for the parameter, while for the activations being

$$\frac{\partial \mathcal{L}}{\partial \mathcal{A}_i} = \frac{\partial \mathcal{L}}{\partial \mathcal{A}_{i+1}} \cdot \frac{\partial \mathcal{A}_{i+1}}{\partial \mathcal{A}_i} = \text{conv}_{\text{full}}\left[\frac{\partial \mathcal{L}}{\partial \mathcal{A}_{i+1}}, \text{rot}(\mathcal{W}_i)\right],$$
$$(2)$$

with $\mathcal{L}$ is the loss computed at the output of the model, $\text{conv}(.)_{\text{full}}$ is the Frobenius inner product, and $\text{rot}(.)$ is a $180°$ rotation operation. It is evident that to compute $\frac{\partial \mathcal{L}}{\partial \mathcal{W}_i}$ and $\frac{\partial \mathcal{L}}{\partial \mathcal{A}_i}$ during the backward pass, $\mathcal{A}_i$ and $\mathcal{W}_i$ must be stored during the forward pass. This storage requirement is indeed the primary cause of memory bottleneck in backpropagation (Lin et al., 2022).

## 3.2. High-Order Singular Value Decomposition for Activation Maps Compression

In this section we recap how activation maps can be decomposed by HOSVD. For each mode $m$, the activation map $\mathcal{A}_i$ can be unfolded into $A_{i,m} \in \mathbb{R}^{a_{i,m} \times b_{i,m}}$, where $(a_{i,m}, b_{i,m}) = \left(\mathcal{D}_{i,m}, \prod_{j, j \neq m} \mathcal{D}_{i,j}\right)$. For instance, in the first mode ($m = 1$), unfolding $\mathcal{A}_i$ results in $A_{i,1} \in \mathbb{R}^{B \times C_i H_i W_i}$.

With optimal ranks $\mathbf{r}_i \in \mathbb{N}^4$ obtained for a given explained variance threshold $\varepsilon \in [0, 1]$, we can perform truncated SVD with the corresponding rank $\mathbf{r}_{i,m} \in \mathbf{r_i}$ on $A_{i,m}$. The final decomposed form reads:

$$\mathcal{A}_i \approx \tilde{\mathcal{S}}_i \times_1 \tilde{U}_{i,1} \times_2 \tilde{U}_{i,2} \times_3 \tilde{U}_{i,3} \times_4 \tilde{U}_{i,4}, \quad (3)$$

where $\tilde{\mathcal{S}}_i \in \mathbb{R}^{\mathbf{r}_{i,1} \times \mathbf{r}_{i,2} \times \mathbf{r}_{i,3} \times \mathbf{r}_{i,4}}$ is the core tensor which can be viewed as a compressed version of $\mathcal{A}_i$, $\tilde{U}_{i,m} \in \mathbb{R}^{a_{i,m} \times \mathbf{r}_{i,m}}$ are the factor matrices and their columns correspond to the principal components over the $m^{th}$ mode. The $m$-mode product "$\times_m$" of a $n^{th}$-order tensor $\mathcal{G} \in \mathbb{R}^{P_1 \times P_2 \times \cdots \times P_n}$ and a matrix $B \in \mathbb{R}^{Q \times P_m}$ is a $n^{th}$-order tensor $\mathcal{R} \in \mathbb{R}^{P_1 \times \cdots \times P_{m-1} \times Q \times P_{m+1} \times \cdots \times P_n}$ which can be expressed as:

$$\mathcal{R}_{p_1, .., p_{m-1}, q, p_{m+1}, .., p_n} =$$
$$\sum_{p_m=1}^{P_m} g_{p_1, .., p_{m-1}, q, p_{m+1}, .., p_n} b_{q, p_m}. \quad (4)$$

Therefore, during the forward pass of a training step, instead of storing $\Theta_{\text{space}}(\prod_{m=1}^{4} \mathcal{D}_{i,m})$ elements of $\mathcal{A}_i$, the application of $\text{HOSVD}_\varepsilon$ reduces this storage requirement to:

$$M_i = \prod_{m=1}^{4} \mathbf{r}_{i,m} + \sum_{m=1}^{4} \mathcal{D}_{i,m} \mathbf{r}_{i,m}. \quad (5)$$

Moreover, the majority of the activation map energy is focused on the first few principal components across all modes (Nguyen et al., 2024). Thus, small values of $\mathbf{r}_{i,m}$ are sufficient to reconstruct $\mathcal{A}_i$ from its low-rank components with acceptably low error.

In the optimal scenario where $\mathbf{r}_{i,m} = 1$, Fig. 2a and Fig. 2b depict the evolution in the total number of FLOPs between vanilla training and $\text{HOSVD}_\varepsilon$ during both the forward and backward passes as the size of the activation map increases. In the backward pass, the gradient can be computed directly on the low-rank components, significantly enhancing computational efficiency compared to vanilla training. The computational speed during the backward pass improves as the rank $\mathbf{r}_{i,m}$ decreases. However, performing $\text{HOSVD}_\varepsilon$ at each training step introduces substantial computational overhead. This overhead grows exponentially with the activation map size, significantly increasing the forward pass FLOPs. The computational savings from low-rank operations in the backward pass cannot compensate for this increased cost. A more detailed analysis can be found in Appendix A.2.

In the next section, we will address the computational overhead problem by leveraging subspace iteration and a custom strategy for activation rank selection at a specific layer's scale.

## 3.3. Subspace Iteration and Rank Selection

To counter the computational complexity introduced by HOSVD, multiple choices could be made, among which subspace iteration (Stewart & Miller, 1975) poses itself as a possible candidate. Subspace iteration is a numerical technique commonly used for computing a subset of the dominant eigenvalues and corresponding eigenvectors of a large matrix (or tensor generalizations). It has already been successfully applied to deep learning in PowerSGD (Vogels et al., 2019) for gradient estimation in data-parallel distributed optimization, showcasing similar performance as singular value decomposition (SVD) but with significantly lower computational cost (Appendix A.1).

The challenge when deploying such a solution for activation map compression will be to estimate the best rank selection, layer-by-layer. Some works are already attempting to attack this problem for parameter compression, passing through a perplexity estimator.

Yuan et al. (2023) introduced Sensitivity-based Truncation Rank Searching (STRS), a method to find adequate truncation rank to compress weights based on a metric called sensitivity, which is derived from the perplexity $\mathcal{P}$. The latter serves as a metric to estimate the approximation error of the model when encountering new data and is defined as follows:

$$\mathcal{P} = e^{\mathcal{L}}, \quad (6)$$

where $\mathcal{L}$ is the loss of the model. Such perplexity can be evaluated layer-by-layer, depending on how much such a layer will be compressed- the goal there will be to compress the most while maintaining $\mathcal{P}$ the lowest. Yuan *et al.* evaluate $\mathcal{P}$ by feeding the model with short input sequences and various compression rates - defined as the ratio between low-rank and full-rank tensor versions. Their analysis reveals that changes in a layer's perplexity are inversely proportional to changes in compression rates, with each layer having its unique proportionality ratio. This ratio is denoted as the layer's sensitivity measure: layers showing larger perplexity increases when compression rates decrease are considered more sensitive.

**Perplexity for activation compression.** In our particular case, compressing the activations only affects the parameters' derivatives and does not affect the output loss of the model. We thus re-define perplexity to activations $\mathcal{P}_{\mathcal{A}_i}$ as the difference between $\frac{\partial \mathcal{L}}{\partial \mathcal{W}_i}$ and $\widetilde{\frac{\partial \mathcal{L}}{\partial \mathcal{W}_i}}$ to better represent the error introduced when performing activation compression.

To estimate perplexity at different compression levels, we wish to perform HOSVD decomposition of four-dimensional tensors, resulting in an exponentially large number of rank combinations across each mode of each layer of the network. To overcome this practical limitation, we replace the set of compression rates with a set $\mathcal{E}$ of explained variance thresholds to create a more equitable metric. We define $\mathcal{E} \in (0, 1]^E$, $E$ corresponding to the number of explained thresholds evaluated, each of which will result in an efficient combination of ranks for all the layers considered. Our proposed perplexity estimation strategy follows two steps.

Step 1. For each explained variance threshold $\varepsilon_j \in \mathcal{E}$, we perform a forward pass with a batch of pre-trained data. At layer $i$, we store both the uncompressed version of the activation map $(\mathcal{A}_i)_j$ and its low-rank variant $(\tilde{\mathcal{A}}_i)_j$, which is obtained by computing HOSVD and truncating the resulting components with respect to $\varepsilon_j$.

Step 2. During the backward pass, $(\frac{\partial \mathcal{L}}{\partial \mathcal{W}_i})_j$ and its corresponding estimated version $(\widetilde{\frac{\partial \mathcal{L}}{\partial \mathcal{W}_i}})_j$ are computed. We define the perplexity $\mathcal{P}_{\mathcal{A}_{i_j}}$ of layer $i$ given $\varepsilon_j$ as the Frobenius norm of the difference between the low-rank and original gradients:

$$\mathcal{P}_{i,j} = \left\| \left( \frac{\partial \mathcal{L}}{\partial \mathcal{W}_i} \right)_j - \left( \widetilde{\frac{\partial \mathcal{L}}{\partial \mathcal{W}_i}} \right)_j \right\|_F. \qquad (7)$$

This process is repeated across all layers, resulting in a perplexity matrix $\mathcal{P} \in \mathbb{R}^{N \times E}$ and a corresponding rank tensor $\mathcal{R}^{N \times E \times 4}$, containing the selected ranks across the 4 modes of activation tensors for each combination of layer and explained variance threshold.

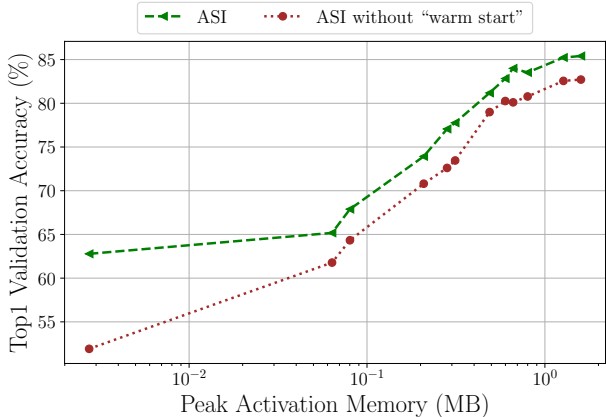

*Figure 3.* Performance of MCUNet pre-trained on ImageNet and finetuned on CIFAR-10 with ASI.

**Rank Selection.** We consider a memory budget constraint $\mathcal{B}$ for the activation memory across the set $\mathcal{F}$ of fine-tuned layers. Our goal here is to optimally select the ranks over each mode while adhering to the memory constraint.

We define a recursive backtracking algorithm to identify the optimal combination of truncation ranks $\mathcal{R}_{\text{opt}}$ based on a set of indices $\mathcal{J} \in \mathbb{N} \cap [1, E]$ across $\mathcal{F}$ such that

$$\mathcal{R}_{\text{opt}_i} = \mathcal{R}_{i,j} \mid j \in \mathcal{J}, \qquad (8)$$

$$\mathcal{J} = \arg\min_{\mathcal{J}} \left( \sum_{i=1}^{|\mathcal{F}|} \sum_{j \in \mathcal{J}} \mathcal{P}_{i,j} \right) \Bigg| \sum_{i=1}^{|\mathcal{F}|} M_i \leq \mathcal{B}, \qquad (9)$$

where $M_i$ corresponds to the resulting activation memory as expressed in (5) with $\mathbf{r}_i \in \mathcal{R}_{\text{opt}}$.

### 3.4. Activation Subspace Iteration (ASI)

Vogels et al. (2019) argue that between each training step, the gradient features only marginally change. Therefore, once properly selected, the projection subspace can be considered stable over a sufficiently small number of steps. Intuitively, due to the linearity of the subspace projection operator, the sequence of steps in the same subspace (warm start) converges to the eigenvector of the averaged stochastic gradients over the same. This results in having lower variance with respect to recalculating the projecting subspace.

When considering such behavior in the context of activation maps, a similar principle applies. Specifically, the activation map $\mathcal{A}_i$ of layer $i$ is computed as follows:

$$\mathcal{A}_i = \sigma \left[ \text{conv} \left( \mathcal{W}_{i-1}, \mathcal{A}_{i-1} \right) \right], \qquad (10)$$

where $\mathcal{W}_{i-1}$ represents the weights of layer $i-1$, and $\sigma(.)$ is the non-linearity function. Assuming that only a small update is made at each step, $\mathcal{W}_{i-1}$ undergoes only minor changes, leading to minor changes in $\mathcal{A}_i$. Thus, over a small

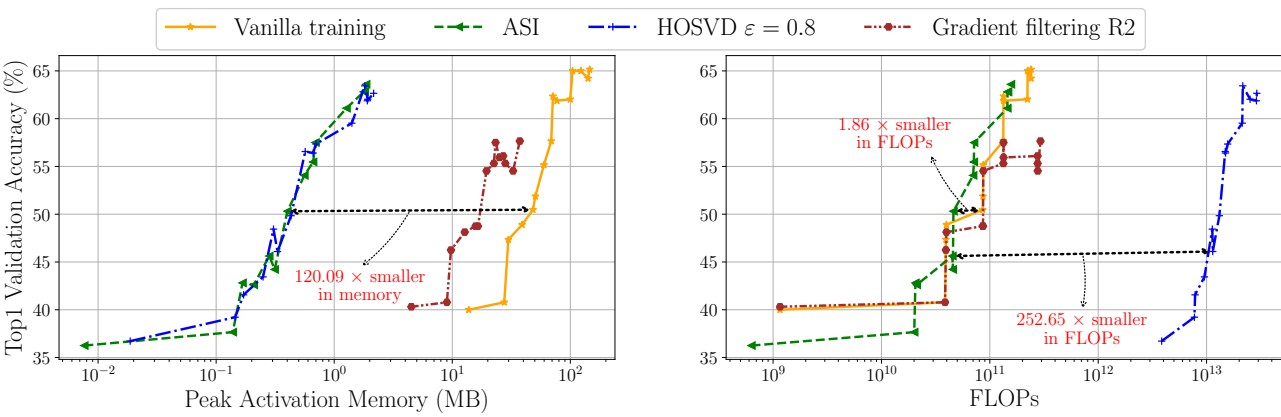

*Figure 4.* Performance of MCUNet pre-trained on ImageNet and finetuned on Pets with different strategies.

---

**Algorithm 1** ASI for layer $i$ with set of rank $\mathbf{r}_i \in \mathcal{R}_{\text{opt}}$

---

**Input:**

Activation map $\mathcal{A}_i^{(t)} \in \mathbb{R}^{B \times C_i \times H_i \times W_i}$ at epoch $t$.
Target ranks for 4 modes $\mathbf{r}_i \in \mathbb{N}^4 \cap [1, \min(a_{i,m}, b_{i,m})]$, where $(a_{i,m}, b_{i,m})$ is shape of activation map $\mathcal{A}_i^{(t)}$ at mode $m$.

**Function:**

Initialize $S_i = \mathcal{A}_i^{(t)}$
**for** $m = 1$ **to** $4$ **do**
$\quad A_{i,m}$ = unfold $\mathcal{A}_i^{(t)}$ along mode $m$ such that $A_{i,m} \in \mathbb{R}^{a_{i,m} \times b_{i,m}}$
$\quad$**if** $t = 0$ **then**
$\quad\quad$ Initialize $V \in \mathbb{R}^{b_{i,m} \times \mathbf{r}_{i,m}}$ from an i.i.d standard normal distribution.
$\quad$**else**
$\quad\quad V = A_{i,m}^T U_{i,m}^{(t)}$
$\quad$**end if**
$\quad U_{i,m}^{(t)} = \text{Orthogonalize}(A_{i,m} V)$
$\quad S_i = S_i \times_m U_{i,m}^{(t)}$
**end for**
**return** $S_i, U_{i,m}^{(t)}$ with $m : 1 \to 4$

---

number of steps, $\mathcal{A}_i$ can also be considered stable. This assumption is further supported by the findings of (Virmaux & Scaman, 2018), where it was observed that most activation functions, such as ReLU, Leaky ReLU, SoftPlus, Tanh, Sigmoid, ArcTan, and Softsign, as well as max-pooling, have a Lipschitz constant equal to 1. Based on this logic, we propose the Activation Subspace Iteration (ASI) method, which applies a single subspace iteration with a "warm start" to effectively estimate the low-rank version of activation maps.

Given optimal ranks $\mathbf{r}_i \in \mathcal{R}_{\text{opt}}$ obtained for a given budget $\mathcal{B}$ following the method proposed in Sec. 3.3, to effectively

produce $\tilde{\mathcal{A}}_i$, we replace the SVD step in the HOSVD$_\varepsilon$ algorithm. In other words, we perform a single subspace iteration with a warm start four times, corresponding to the four modes of the activation map. Algorithmic details are provided in Algorithm 1.

### 3.5. Computational Complexity

In this section, we assess the computational complexity of the compression operations during the training process. Perplexity search and rank selection are performed offline and only once for each model; therefore, we exclude them from this evaluation.

Following the notation from previous sections, the computational overhead of our method, $O_{\text{ASI}}$ is given by (14). HOSVD$_\varepsilon$ incurs the following overhead:

$$\text{O}_{\text{HOSVD}_\varepsilon} = \sum_{d \in \mathcal{D}_i} \max\left(d, P_d\right)^2 \times \min\left(d, P_d\right), \quad (11)$$

where $P_d = \prod_{d' \in \mathcal{D}_i \setminus \{d\}} d'$ (Nguyen et al., 2024). This is even more significant since the first few singular values in each mode typically carry most of the energy of the entire activation map. In other terms, the target rank $\mathbf{r}_i$ obtained as an output of the rank selection algorithm presented in Sec. 3.3, is expected to be low compared to the full dimension, resulting in considerable compression ratios with limited information loss. Fig. 2d illustrates the speedup ratio $R_S$ between vanilla training and ASI, defined as their FLOPs ratio. Our method achieves faster training compared to vanilla training when considering smaller values of truncation ranks $\mathbf{r}_i$ and larger activation maps, while reaching similar rates of memory savings as HOSVD$_\varepsilon$ (Fig. 2c). Detailed analysis can be found in Appendix A.2 and A.3.

## 4. Experiments

In this section, we present experiments that demonstrate the effectiveness of our proposed method. First, we provide,

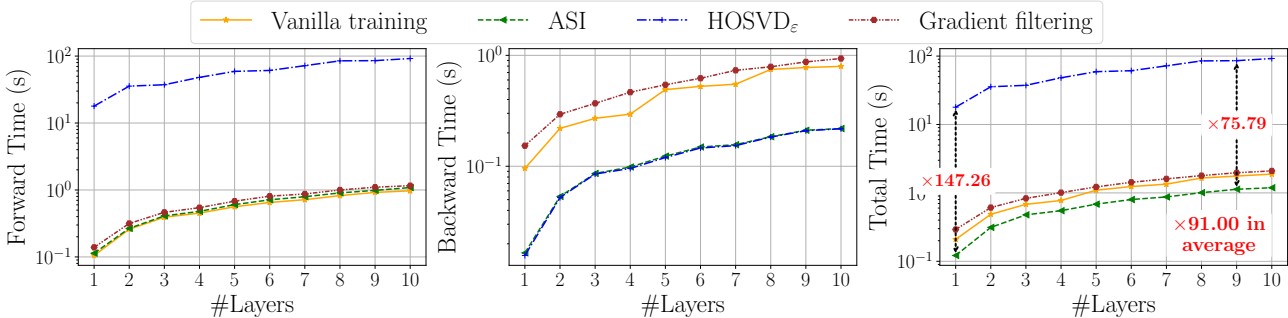

Figure 5. Training time of MCUNet over 5 iterations on CIFAR-10 with batch size 128 on a Raspberry Pi 5.

in Sec. 4.1, an overview of the experimental setup. Then, in Sec. 4.2, we conduct an ablation study to analyze the impact of the warm-start process on the performance of ASI. Sec. 4.3 reports the results obtained when applying ASI to state-of-the-art architectures and datasets. Finally, we compare the techniques in a real-world deployment setting(Sec. 4.4). All simulation experiments are conducted using PyTorch 1.13.1 on an NVIDIA Quadro RTX A4500 with 20GB of VRAM, while the on-device experiments are performed on a Raspberry Pi 5 with a Cortex-A76 CPU and 8GB RAM.

### 4.1. Experimental Setup

**Overview.** To evaluate the effectiveness of ASI, we conduct classification tasks on six datasets: CIFAR-10, CIFAR-100 (Krizhevsky, 2009), CUB (Wah et al., 2011), Flowers (Nilsback & Zisserman, 2008), Pets (Zhang et al., 2022), and ImageNet (Deng et al., 2009). Since practical feasibility is a key concern, most experiments are performed using MCUNet (Lin et al., 2022), a model that has been successfully deployed on real edge devices with only 256kB of memory. Additionally, we extend our evaluation to other compact architectures, including ResNet-18, ResNet-34 (He et al., 2016), and MobileNetV2 (Sandler et al., 2018). Detailed setup information can be found in Appendix B.1.

**Rank Selection.** We employ a set of explained variance thresholds $\mathcal{E} = \{0.4, 0.5, 0.6, 0.7, 0.8, 0.9\}$ to measure the perplexity $\mathcal{P}$ of each layer.

**Memory Budget.** Since HOSVD$_\varepsilon$ operates under an explained variance constraint without strict memory control, it does not enforce an explicit memory budget. To ensure a fair comparison, we set the peak memory consumption of HOSVD$_\varepsilon$ as the memory budget for ASI when fine-tuning the same number of layers.

**Result Logging.** The reported results include FLOPs and memory usage for both ASI and vanilla training. For HOSVD$_\varepsilon$, we report peak FLOPs and peak memory, as its resource consumption varies dynamically, unlike ASI

and vanilla training, which remain constant.

### 4.2. Ablation Study

Concerning this experiment, we evaluate the impact of warm-start on ASI when fine-tuning MCUNet at different depths. The model is pre-trained on ImageNet-1k, with CIFAR-10 as the downstream dataset. The number of fine-tuned layers gradually increases from the final layer upward, where the first marker in each plot indicates the case in which only the last layer is fine-tuned. Fig. 3 presents the experimental results, showing that even when warm-start is not applied—equivalent to assuming that activation maps across layers are entirely independent—the model still converges with just a single subspace iteration. On average, incorporating "warm-start" improves ASI accuracy by 3.87%.

### 4.3. Main results

**MCUNet with Pets.** We conduct a similar experiment as in the previous section, using the Pets dataset as the downstream task to compare the performance of ASI, HOSVD$_\varepsilon$, and vanilla training. Following the findings of Nguyen *et al.*, we set $\varepsilon = 0.8$ to achieve a good compression ratio while minimizing information loss with HOSVD$_\varepsilon$. The peak memory usage of HOSVD$_\varepsilon$ is then used as the memory budget constraint for ASI. The experimental results are illustrated in Fig. 4, where it is evident that ASI outperforms both others. It achieves a compression ratio comparable to HOSVD$_\varepsilon$ while consuming even fewer FLOPs than vanilla training for the same number of fine-tuned layers.

Notably, at a similar accuracy level, ASI reduces memory usage by up to $120.09\times$ compared to vanilla training. Additionally, ASI requires $252.65\times$ fewer FLOPs than HOSVD. Overall, ASI achieves comparable accuracy to that of HOSVD$_\varepsilon$ while saving up to $1.86\times$ of the computational cost. We also performed experiments on other downstream tasks, included in Appendix B.3.

**ImageNet-1k Classification.** Similar experiments with Im-

*Table 1.* Experimental results on ImageNet with "#Layers" denotes the number of fine-tuned convolutional layers, counted from the model's end. Activation memory is presented in MegaBytes (MB), and computational complexity is expressed in terms of Gigaflops (GFLOPs). Gradient filtering is utilized with patch size R2.

| Method | MobileNetV2 | | | Method | ResNet18 | | | |
|---|---|---|---|---|---|---|---|---|
| | #Layers | Acc ↑ | Mem (MB) ↓ | GFLOPs ↓ | | #Layers | Acc ↑ | Mem (MB) ↓ | GFLOPs ↓ |
| Vanilla training | All | 74.0 | 1651.84 | 830.82 | Vanilla training | All | 72.8 | 532.88 | 291.79 |
| | 2 | 62.6 | 15.31 | 4.50 | | 2 | 69.9 | 12.25 | 29.60 |
| | 4 | 65.8 | 28.71 | 57.48 | | 4 | 71.5 | 30.63 | 46.45 |
| Gradient filtering R2 | 2 | 62.6 | 5.00 | 4.50 | Gradient filtering R2 | 2 | 68.7 | 4.00 | 29.60 |
| | 4 | 65.2 | 9.38 | 57.50 | | 4 | 69.3 | 7.00 | 47.70 |
| HOSVD ($\varepsilon = 0.8$) | 2 | 61.1 | 0.15 | 3049.71 | HOSVD ($\varepsilon = 0.8$) | 2 | 69.2 | 0.97 | 1581.42 |
| | 4 | 63.9 | 0.73 | 5895.31 | | 4 | 70.5 | 2.89 | 4048.56 |
| ASI | 2 | 60.3 | 0.13 | 2.81 | ASI | 2 | 68.9 | 0.93 | 19.04 |
| | 4 | 64.0 | 0.71 | 31.54 | | 4 | 70.6 | 2.63 | 33.14 |

| Method | MCUNet | | | Method | ResNet34 | | | |
|---|---|---|---|---|---|---|---|---|
| | #Layers | Acc ↑ | Mem (MB) ↓ | GFLOPs ↓ | | #Layers | Acc ↑ | Mem (MB) ↓ | GFLOPs ↓ |
| Vanilla training | All | 67.4 | 632.98 | 248.84 | Vanilla training | All | 75.6 | 839.04 | 528.55 |
| | 2 | 62.1 | 13.78 | 19.31 | | 2 | 69.6 | 12.25 | 29.60 |
| | 4 | 64.7 | 19.52 | 19.88 | | 4 | 72.2 | 24.50 | 59.19 |
| Gradient filtering R2 | 2 | 61.8 | 4.50 | 19.31 | Gradient filtering R2 | 2 | 68.8 | 4.00 | 29.60 |
| | 4 | 64.4 | 6.38 | 19.89 | | 4 | 70.9 | 8.00 | 59.21 |
| HOSVD ($\varepsilon = 0.8$) | 2 | 61.7 | 0.48 | 1988.05 | HOSVD ($\varepsilon = 0.8$) | 2 | 68.7 | 0.30 | 1579.64 |
| | 4 | 63.9 | 0.88 | 2457.59 | | 4 | 71.1 | 1.11 | 3160.46 |
| ASI | 2 | 61.7 | 0.38 | 11.01 | ASI | 2 | 68.3 | 0.25 | 16.44 |
| | 4 | 63.5 | 0.83 | 11.93 | | 4 | 71.1 | 1.09 | 35.31 |

ageNet were conducted across different models. The results are shown in Table 1. We observe that, in all cases, for the same depth, ASI consistently consumes the least resources while achieving accuracy comparable to $\text{HOSVD}_\varepsilon$.

The memory consumption of $\text{HOSVD}_\varepsilon$ is not uniform—it tends to be lower at the beginning of training, then surges to a peak before slightly decreasing as the model converges. Intuitively, the actual memory requirement of $\text{HOSVD}_\varepsilon$ does not necessarily correspond to its peak memory usage but rather a lower value. This explains why, despite using $\text{HOSVD}_\varepsilon$'s memory consumption as the budget, ASI never actually reaches that threshold in practice.

### 4.4. On-device Latency

We conduct similar experiments in a real embedded environment, using a Raspberry Pi 5. We report the estimations over the first 5 iterations of MCUNet trained on CIFAR-10, using batch size 128. The results of our experiment are shown in Fig. 5. During the forward pass, due to the high computational complexity of performing HOSVD at each training step, $\text{HOSVD}_\varepsilon$ takes, on average, $106.13\times$ longer than both ASI and vanilla training. In the backward pass, the low-rank computation enables ASI and HOSVD to be $3.95\times$ faster than vanilla training. This speed advantage is

sufficient to compensate for the minimal overhead of ASI, making it $91.0\times$ faster than HOSVD, $1.86\times$ faster than gradient filtering, and $1.56\times$ faster than vanilla training.

## 5. Conclusion

In this work we introduced ASI (Activation Subspace Iteration) as a shortcut approach to address the activation memory bottleneck during backpropagation, enabling the feasibility of on-device learning. Taking advantage of the short-term stability of activation maps during training, the combination of a single subspace iteration and reusing previous approximations enables us to achieve up to $120.09\times$ activation memory reduction and up to $1.86\times$ fewer training FLOPs while maintaining similar performances as vanilla training. Additionally, our method delivers strong improvements in latency, achieving up to a $91.0\times$ times speedup when deployed on a Raspberry Pi 5, compared to state-of-the-art approaches. In the future, we wish to explore extending ASI to larger models, such as large language models, and further optimizing its computational aspects to broaden its applicability.

## Acknowledgements

Part of this work was funded by Hi!PARIS Center on Data Analytics and Artificial Intelligence, by the European Union's Horizon Europe Research and Innovation Programme under grant agreement No. 101120237 (ELIAS), by the European Union's HORIZON Research and Innovation Programme under grant agreement No 101120657, project ENFIELD (European Lighthouse to Manifest Trustworthy and Green AI) and by French National Research Agency (ANR-22-PEFT-0003 and ANR-22-PEFT-0007) as part of France 2030, the NF-NAI project and NF-FITNESS project.

## Impact Statement

This paper presents work whose goal is to advance the field of Machine Learning. There are many potential societal consequences of our work, none which we feel must be specifically highlighted here.

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

# A. Additional Theoretical Details

Following the notation introduced in the main paper, we propose in this section to analitically study the overhead introduced in the forward pass when performing each compression algorithm, the speedup of efficient weight derivative computation and the required space to store the compressed activations.

## A.1. Subspace Iteration

In this section, we present the subspace iteration technique from (Vogels et al., 2019) for compressing activation maps instead of gradients. The core idea remains unchanged: a single step of subspace iteration (Stewart & Miller, 1975) is employed to obtain a fast low-rank approximation of the given matrix. In our case, this matrix corresponds to the four unfolded versions of the activation maps along their respective modes. However, as in the original work, a single step of subspace iteration typically does not provide a sufficiently accurate low-rank approximation. To mitigate this, Vogels *et al.* suggest reusing the low-rank approximation from the previous iteration as the initialization for the current iteration.

This reuse is particularly well-motivated in our setting. Although activation maps evolve as network parameters are updated, their changes between consecutive iterations are relatively small. This behavior stems from the Lipschitz continuity of activation functions (Virmaux & Scaman, 2018) and the "tiny" incremental nature of parameter updates during optimization. By leveraging the previous approximation, we effectively smooth the sequence of activation maps across iterations, reducing the variance of the low-rank approximation compared to the case without reuse. A formal proof of this property can be found in (Vogels et al., 2019).

**Overhead.** Consider a model's gradient matrix $M^{(t)} \in \mathbb{R}^{a \times b}$ at time $t$ with a targeted rank $r \in [1, \min(a,b)] \cap \mathbb{N}$. Algorithm 2 requires performing two matrix multiplications $P^{(t)} = M^{(t)}Q^{(t)}$ and $Q^{(t)} = {M^{(t)}}^T P^{(t)}$, which cost $\Theta_{\text{FLOPs}}(2abr)$, and an orthogonalization using Gram-Schmidt that costs $\Theta_{\text{FLOPs}}(r^3)$. Therefore, the total overhead FLOPS is:

$$O_{\text{SIW}} = 2abr + r^3 \tag{12}$$

---

**Algorithm 2** Subspace iteration with Warm start at epoch $t$

---

**Input:**
  Epoch index $t$
  Matrix $M^{(t)} \in \mathbb{R}^{a \times b}$
  Targeted rank $r \in [1, \min(a,b)] \cap \mathbb{N}$
**Function:**
  **if** $t = 0$ **then**
      Initialize $Q^{(t)} \in \mathbb{R}^{b \times r}$ from an i.i.d standard normal distribution.
  **else**
      $Q^{(t)} = Q_{(t-1)}$ # This is "Warm start"
  **end if**
  $P^{(t)} = M^{(t)}Q^{(t)}$
  $\hat{P}^{(t)} = \text{Orthogonalize}(P^{(t)})$
  $Q^{(t)} = {M^{(t)}}^T \hat{P}^{(t)}$
  **return** $\hat{P}^{(t)}, Q^{(t)}$

---

## A.2. HOSVD Overhead

As proved by Nguyen et al. (2024), performing HOSVD at each training iteration yeilds to decompose an activation map $\mathcal{A}_i \in \mathbb{R}^{B_i, C_i, H_i, W_i}$ with $\mathcal{D}_i = \{B_i, C_i, H_i, W_i\}$ yields:

$$O_{\text{HOSVD}_\varepsilon} = \sum_{d \in \mathcal{D}_i} \max(d, P_d)^2 \times \min(d, P_d), \tag{13}$$

where $P_d = \prod_{d' \in \mathcal{D}_i \setminus \{d\}} d'$.

### A.3. Details of Overhead, Computational Speedup and Space Complexity between ASI and Vanilla Training

**Overhead.** Using ASI to decompose the tensor is equivalent to performing subspace iteration on each of its mode. Considering mode $m$, decomposing the unfolded version $A_{i,m} \in \mathbb{R}^{d \times d'}$ of $\mathcal{A}_i$ requires a cost of $\Theta_{\text{FLOPs}}(2dd'\mathbf{r}_{i,m} + \mathbf{r}_{i,m}^3)$, where $d$ and $d'$ represent the dimensions of $A_{i,m}$. Consequently, performing subspace iteration across all four modes incurs a total cost of:

$$O_{\text{ASI}} = \sum_{m=1}^{4} \left(2dd'\mathbf{r}_{i,m} + \mathbf{r}_{i,m}^3\right) \mid d = \mathcal{D}_{i,m}, d' = \mathcal{D} \setminus \{d\}. \tag{14}$$

**Speedup.** Leveraging the low-rank gradient computation technique in Nguyen et al. (2024), the backward pass at a training step of ASI will cost:

$$C_{\text{ASI}} = \mathbf{r}_{i,1}B_iC_i'H_i'W_i' + \mathbf{r}_{i,1}\mathbf{r}_{i,2}\mathbf{r}_{i,3}\mathbf{r}_{i,4}H_i + \mathbf{r}_{i,1}\mathbf{r}_{i,2}\mathbf{r}_{i,4}H_iW_i + \mathbf{r}_{i,1}\mathbf{r}_{i,2}C_i'H_i'W_i'D_i^2 + \mathbf{r}_{i,2}C_i'C_iD_i^2 \tag{15}$$

While this cost of vanilla training is:

$$C_{\text{vanilla}} = D_i^2C_iC_i'B_iH_i'W_i' \tag{16}$$

Moreover, the amount of FLOPs that vanilla training requires to perform the forward pass is:

$$O_{\text{vanilla}} = D_i^2C_iC_i'B_iH_iW_i \tag{17}$$

We then can deduce the speedup ratio $R_S$ for a training step between vanilla training and ASI as follow:

$$R_S = \frac{O_{\text{vanilla}} + C_{\text{vanilla}}}{O_{\text{vanilla}} + O_{\text{ASI}} + C_{\text{ASI}}} \tag{18}$$

**Space complexity.** Essentially, ASI performs compression in a manner similar to $\text{HOSVD}_\varepsilon$, achieving a comparable compression ratio. As a result, the space complexity ratio (i.e., the memory ratio between vanilla training and ASI) is the same as $\text{HOSVD}_\varepsilon$, which is:

$$R_C = \frac{B_iC_iH_iW_i}{\mathbf{r}_{i,1}\mathbf{r}_{i,2}\mathbf{r}_{i,3}\mathbf{r}_{i,4} + B\mathbf{r}_{i,1} + C\mathbf{r}_{i,2} + H\mathbf{r}_{i,3} + W\mathbf{r}_{i,4}}. \tag{19}$$

## B. Additional Experimental Details

### B.1. Detailed Experiment Setup

To ensure a fair comparison, we adopt the same experimental setup as Nguyen et al. (2024), with the details outlined below:

**ImageNet.** The dataset is divided into two equal-sized, non-i.i.d. partitions using FedAvg (McMahan et al., 2017). Models are pretrained on the first partition, while the second partition is used for fine-tuning, with $80\%$ allocated for training and the remaining $20\%$ as the validation set. We fine-tune the provided checkpoints for 90 epochs, applying L2 gradient clipping with a threshold of 2.0. Optimization is performed using SGD with a weight decay of $1 \times 10^{-4}$ and a momentum of $0.9$. Data augmentation techniques include random resizing, flipping, normalization, and mini-batching with a size of 64. The loss function is cross-entropy. The learning rate increases linearly over four warm-up epochs, reaching $0.005$, and then follows a cosine annealing decay schedule.
**Other Datasets.** The models are first pretrained on ImageNet before being fine-tuned on a completely different downstream dataset with the same training-validation ratio as above. Training is conducted using cross-entropy loss with the SGD optimizer. The initial learning rate is set to $0.05$ and follows a cosine annealing decay schedule. Momentum is fixed at $0$, and the weight decay remains at $1 \times 10^{-4}$. Additionally, L2 gradient clipping with a threshold of 2.0 is applied.

**Semantic Segmentation.** Following the training policy described in Yang et al. (2023), we used the pretrained and calibrated checkpoints provided by the authors. These models were first pretrained on the Cityscapes dataset using MMSegmentation, and then fine-tuned on the augmented Pascal VOC12 dataset. We used a learning rate that starts at $0.01$ and follows a cosine annealing schedule. The weight decay was set to $5 \times 10^{-4}$ and momentum to $0.9$. We trained with a batch size of 8. For data augmentation, we applied random cropping, horizontal flipping, photometric distortions, and image normalization. The models were fine-tuned for a total of $20,000$ steps using cross-entropy loss.

### B.2. Perplexity with different values of $\varepsilon$

Fig. 6 illustrates the change in perplexity of the last four layers of MCUNet when evaluated with different explained variance thresholds $\varepsilon$. As expected, higher values of $\varepsilon$ result in lower perplexity. However, when $\varepsilon$ decreases below 0.5, there is no noticeable change in perplexity. This phenomenon arises because most of the energy in the activation maps is concentrated in the first singular value, as demonstrated by Nguyen et al. (2024); in this case, the first singular value alone captures more than 50% of the total energy. Therefore, experimenting with $\varepsilon$ smaller than this value does not lead to any further improvement. To save computational resources, we limit our perplexity measurements to $\varepsilon > 0.4$ in all experiments.

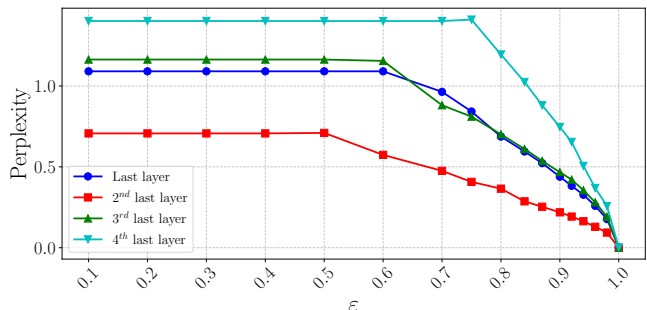

*Figure 6.* Perplexity as a function of the explained variance threshold $\varepsilon$ for the last layers of MCUNet.

### B.3. Additional Results

**Additional Classification Results.** Following the same training policy described in Sec. 4.3, we compare the performance of various techniques across different models, all of which are pretrained on ImageNet-1K. The results, presented in Table 2, demonstrate that ASI consistently achieves a comparable compression rate to HOSVD$_\varepsilon$ while maintaining similar accuracy. Furthermore, ASI yields a substantial reduction in FLOPs, even outperforming vanilla training in terms of computational efficiency.

**Semantic Segmentation.** Table 3 reports the performance of the evaluated techniques on the segmentation task. As previously observed, ASI continues to demonstrate superior performance in terms of both compression ratio and computational cost, confirming its effectiveness beyond classification.

**ASI on Large Language Models.** In this experiment, we fine-tune TinyLlama 1B (Zhang et al., 2024) on the BoolQ (Clark et al., 2019) dataset to compare ASI with vanilla training. The dataset is divided into batches of size 8, with each sample having a maximum length of 512 tokens. Due to computational cost, applying HOSVD$_\varepsilon$ to this model is infeasible. Therefore, instead of budget-based compression, we directly set the compression rank to 20. Table 4 presents the results when fine-tuning from 1 and 5 layers. Notably, fine-tuning 5 layers leads to up to a 2500× reduction in activation memory, and computational cost decreases by approximately 1.9×, while retaining decent accuracy. Remarkably, as the number of fine-tuned layers increases, both memory compression and FLOP savings improve significantly under ASI.

*Table 2.* More classification results

| Model | Method | #Layers | CUB200 | | | Flowers102 | | | Pets | | | CIFAR-10 | | | CIFAR-100 | | |
|---|---|---|---|---|---|---|---|---|---|---|---|---|---|---|---|---|---|
| | | | Acc ↑ | Mem (MB)↓ | TFLOPs↓ | Acc ↑ | Mem (MB)↓ | TFLOPs↓ | Acc ↑ | Mem (MB)↓ | TFLOPs↓ | Acc ↑ | Mem (MB)↓ | TFLOPs↓ | Acc ↑ | Mem (MB)↓ | TFLOPs↓ |
| MobileNet V2 | Vanilla training | All | 54.25 | 3303.67 | 1.66 | 80.58 | 3303.67 | 1.66 | 89.84 | 3303.67 | 1.66 | 95.08 | 3303.67 | 1.66 | 78.03 | 3303.67 | 1.66 |
| | | 2 | 49.57 | 30.63 | 0.01 | 81.25 | 30.63 | 0.01 | 88.28 | 30.63 | 0.01 | 88.03 | 30.63 | 0.01 | 64.78 | 30.63 | 0.01 |
| | | 4 | 53.73 | 57.42 | 0.11 | 83.15 | 57.42 | 0.11 | 89.53 | 57.42 | 0.11 | 89.45 | 57.42 | 0.11 | 67.57 | 57.42 | 0.11 |
| | Gradient filtering R2 | 2 | 46.68 | 10.00 | 0.01 | 80.92 | 10.00 | 0.01 | 88.28 | 10.00 | 0.01 | 87.90 | 10.00 | 0.01 | 64.72 | 10.00 | 0.01 |
| | | 4 | 50.20 | 18.75 | 0.11 | 82.70 | 18.75 | 0.11 | 89.22 | 18.75 | 0.11 | 89.25 | 18.75 | 0.11 | 67.22 | 18.75 | 0.11 |
| | HOSVD ($\varepsilon = 0.8$) | 2 | 46.27 | 0.27 | 11.88 | 79.02 | 0.30 | 11.88 | 88.28 | 0.21 | 11.88 | 85.65 | 0.26 | 11.88 | 61.17 | 0.16 | 11.88 |
| | | 4 | 50.61 | 0.67 | 22.91 | 81.58 | 0.76 | 22.92 | 89.69 | 0.77 | 22.92 | 88.48 | 0.68 | 22.92 | 66.09 | 0.71 | 22.92 |
| | ASI | 2 | 46.44 | 0.26 | 0.01 | 80.13 | 0.26 | 0.01 | 88.13 | 0.15 | 0.01 | 85.09 | 0.26 | 0.01 | 61.03 | 0.15 | 0.01 |
| | | 4 | 50.69 | 0.63 | 0.06 | 81.58 | 0.63 | 0.06 | 88.91 | 0.76 | 0.06 | 88.03 | 0.63 | 0.06 | 66.18 | 0.63 | 0.06 |
| MCUNet | Vanilla training | All | 31.94 | 1265.96 | 0.50 | 44.53 | 1265.96 | 0.50 | 74.37 | 1265.96 | 0.50 | 91.25 | 1265.96 | 0.50 | 65.11 | 1265.96 | 0.50 |
| | | 2 | 11.46 | 27.56 | 0.04 | 39.29 | 27.56 | 0.04 | 40.78 | 27.56 | 0.04 | 72.20 | 27.56 | 0.04 | 45.49 | 27.56 | 0.04 |
| | | 4 | 12.59 | 39.05 | 0.04 | 43.42 | 39.05 | 0.04 | 48.91 | 39.05 | 0.04 | 83.53 | 39.05 | 0.04 | 54.93 | 39.05 | 0.04 |
| | Gradient filtering R2 | 2 | 9.96 | 9.00 | 0.04 | 39.06 | 9.00 | 0.04 | 40.47 | 9.00 | 0.04 | 71.18 | 9.00 | 0.04 | 45.07 | 9.00 | 0.04 |
| | | 4 | 12.30 | 12.75 | 0.04 | 42.63 | 12.75 | 0.04 | 49.06 | 12.75 | 0.04 | 82.51 | 12.75 | 0.04 | 54.25 | 12.75 | 0.04 |
| | HOSVD ($\varepsilon = 0.8$) | 2 | 9.20 | 0.13 | 7.73 | 34.82 | 0.14 | 7.73 | 39.22 | 0.14 | 7.73 | 65.48 | 0.07 | 7.73 | 41.18 | 0.05 | 7.73 |
| | | 4 | 8.07 | 0.26 | 9.56 | 31.03 | 0.13 | 9.56 | 43.44 | 0.25 | 9.56 | 77.29 | 0.25 | 9.56 | 49.71 | 0.19 | 9.56 |
| | ASI | 2 | 9.20 | 0.06 | 0.02 | 35.49 | 0.13 | 0.02 | 37.66 | 0.14 | 0.02 | 65.17 | 0.06 | 0.02 | 40.70 | 0.04 | 0.02 |
| | | 4 | 8.16 | 0.21 | 0.02 | 33.04 | 0.11 | 0.02 | 42.81 | 0.17 | 0.02 | 73.93 | 0.21 | 0.02 | 42.34 | 0.19 | 0.02 |
| ResNet18 | Vanilla training | All | 55.12 | 1065.75 | 0.58 | 81.81 | 1065.75 | 0.58 | 89.53 | 1065.75 | 0.58 | 95.32 | 1065.75 | 0.58 | 78.91 | 1065.75 | 0.58 |
| | | 2 | 57.20 | 24.50 | 0.06 | 84.15 | 24.50 | 0.06 | 89.22 | 24.50 | 0.06 | 91.13 | 24.50 | 0.06 | 70.51 | 24.50 | 0.06 |
| | | 4 | 55.30 | 61.25 | 0.09 | 84.60 | 61.25 | 0.09 | 88.75 | 61.25 | 0.09 | 92.43 | 61.25 | 0.09 | 73.33 | 61.25 | 0.09 |
| | Gradient filtering R2 | 2 | 55.08 | 8.00 | 0.06 | 83.71 | 8.00 | 0.06 | 88.75 | 8.00 | 0.06 | 90.43 | 8.00 | 0.06 | 69.80 | 8.00 | 0.06 |
| | | 4 | 50.20 | 14.00 | 0.10 | 83.48 | 14.00 | 0.10 | 88.59 | 14.00 | 0.10 | 91.55 | 14.00 | 0.10 | 71.62 | 14.00 | 0.10 |
| | HOSVD ($\varepsilon = 0.8$) | 2 | 56.86 | 1.75 | 6.13 | 82.14 | 1.59 | 6.13 | 88.75 | 2.15 | 6.13 | 90.80 | 1.51 | 6.13 | 70.09 | 1.15 | 6.13 |
| | | 4 | 55.38 | 3.51 | 15.57 | 83.71 | 3.82 | 15.57 | 88.75 | 4.43 | 15.57 | 91.87 | 1.92 | 15.57 | 71.71 | 1.62 | 15.57 |
| | ASI | 2 | 56.77 | 1.51 | 0.04 | 82.59 | 1.51 | 0.04 | 88.75 | 2.01 | 0.04 | 90.77 | 1.24 | 0.04 | 69.70 | 0.90 | 0.04 |
| | | 4 | 55.73 | 3.50 | 0.07 | 84.82 | 3.75 | 0.07 | 88.44 | 4.34 | 0.07 | 91.90 | 1.89 | 0.06 | 71.93 | 1.61 | 0.06 |
| ResNet34 | Vanilla training | All | 60.42 | 1678.25 | 1.06 | 77.90 | 1678.25 | 1.06 | 90.47 | 1678.25 | 1.06 | 96.71 | 1678.25 | 1.06 | 81.82 | 1678.25 | 1.06 |
| | | 2 | 59.46 | 24.50 | 0.06 | 83.15 | 24.50 | 0.06 | 90.94 | 24.50 | 0.06 | 91.15 | 24.50 | 0.06 | 70.46 | 24.50 | 0.06 |
| | | 4 | 60.59 | 49.00 | 0.12 | 83.15 | 49.00 | 0.12 | 90.31 | 49.00 | 0.12 | 92.43 | 49.00 | 0.12 | 72.83 | 49.00 | 0.12 |
| | Gradient filtering R2 | 2 | 57.52 | 8.00 | 0.06 | 81.14 | 8.00 | 0.06 | 90.94 | 8.00 | 0.06 | 90.52 | 8.00 | 0.06 | 70.03 | 8.00 | 0.06 |
| | | 4 | 58.11 | 16.00 | 0.12 | 82.14 | 16.00 | 0.12 | 90.16 | 16.00 | 0.12 | 91.59 | 16.00 | 0.12 | 70.57 | 16.00 | 0.12 |
| | HOSVD ($\varepsilon = 0.8$) | 2 | 58.77 | 0.80 | 6.13 | 80.02 | 0.69 | 6.13 | 90.63 | 0.92 | 6.13 | 90.61 | 0.56 | 6.13 | 69.86 | 0.44 | 6.13 |
| | | 4 | 58.77 | 1.70 | 12.25 | 81.70 | 1.48 | 12.25 | 90.16 | 2.16 | 12.26 | 91.53 | 1.25 | 12.25 | 70.68 | 1.02 | 12.25 |
| | ASI | 2 | 58.77 | 0.60 | 0.03 | 80.36 | 0.60 | 0.03 | 91.09 | 0.81 | 0.04 | 90.09 | 0.49 | 0.03 | 69.66 | 0.44 | 0.03 |
| | | 4 | 58.85 | 1.58 | 0.07 | 81.70 | 1.38 | 0.07 | 91.41 | 2.05 | 0.07 | 91.54 | 1.24 | 0.07 | 70.60 | 1.00 | 0.07 |
| SwinT | Vanilla training | All | 75.87 | 3491.25 | 1.40 | 93.08 | 3491.25 | 1.40 | 94.06 | 3491.25 | 1.40 | 98.00 | 3491.25 | 1.40 | 87.48 | 3491.25 | 1.40 |
| | | 2 | 72.31 | 91.88 | 0.11 | 86.27 | 91.88 | 0.11 | 93.75 | 91.88 | 0.11 | 94.23 | 91.88 | 0.11 | 77.93 | 91.88 | 0.11 |
| | | 4 | 76.82 | 183.75 | 0.23 | 89.29 | 183.75 | 0.23 | 93.91 | 183.75 | 0.23 | 95.22 | 183.75 | 0.23 | 80.85 | 183.75 | 0.23 |
| | HOSVD ($\varepsilon = 0.8$) | 2 | 70.57 | 2.76 | 116.01 | 85.38 | 3.83 | 116.01 | 93.44 | 4.77 | 116.01 | 93.57 | 6.36 | 116.01 | 77.13 | 5.07 | 116.01 |
| | | 4 | 76.65 | 4.22 | 232.01 | 89.51 | 6.23 | 232.02 | 94.38 | 7.99 | 232.01 | 94.99 | 11.68 | 232.02 | 80.19 | 9.97 | 232.02 |
| | ASI | 2 | 70.66 | 2.45 | 0.08 | 85.71 | 3.54 | 0.09 | 93.44 | 3.54 | 0.09 | 93.83 | 5.96 | 0.09 | 77.11 | 4.88 | 0.09 |
| | | 4 | 77.78 | 3.45 | 0.15 | 88.95 | 5.89 | 0.17 | 93.91 | 7.98 | 0.17 | 95.15 | 11.21 | 0.18 | 80.87 | 9.87 | 0.18 |

# C. Limitation

In essence, the backtracking algorithm is a brute-force method. As the number of layers requiring fine-tuning increases, the size of $\mathcal{P}$ grows accordingly. Solving the optimization problem (9) becomes highly resource-intensive, even though it can be performed offline on a powerful server. More efficient search algorithms need to be developed to replace this approach.

Table 3. Experimental results for semantic segmentation. mIoU is the mean Intersection over Union, and mAcc is the micro averaged accuracy.

| Method | #Layers | mIoU ↑ | mAcc ↑ | Mem (MB) ↓ | TFLOPs ↓ | Method | #Layers | mIoU ↑ | mAcc ↑ | Mem (MB) ↓ | TFLOPs ↓ |
|---|---|---|---|---|---|---|---|---|---|---|---|
| | | **PSPNet (Zhao et al., 2017)** | | | | | | **PSPNet-M (Zhao et al., 2017)** | | | |
| Vanilla training | All | 54.97 | 68.46 | 920.78 | 0.88 | Vanilla training | All | 48.92 | 62.11 | 2622.49 | 2.96 |
| | 5 | 39.36 | 51.79 | 128.00 | 0.08 | | 5 | 36.22 | 46.31 | 104.00 | 0.06 |
| | 10 | 53.17 | 67.18 | 352.00 | 0.62 | | 10 | 45.62 | 58.35 | 604.00 | 1.19 |
| Gradient Filter | 5 | 39.34 | 51.59 | 8.00 | 0.08 | Gradient Filter | 5 | 35.73 | 45.78 | 6.50 | 0.06 |
| | 10 | 51.20 | 65.01 | 22.00 | 0.62 | | 10 | 44.89 | 57.38 | 37.75 | 1.19 |
| HOSVD ($\varepsilon = 0.8$) | 5 | 38.11 | 49.29 | 0.47 | 177.06 | HOSVD ($\varepsilon = 0.8$) | 5 | 33.40 | 42.50 | 0.03 | 117.06 |
| | 10 | 49.23 | 62.39 | 1.40 | 322.25 | | 10 | 40.06 | 51.79 | 1.47 | 744.71 |
| ASI | 5 | 37.90 | 49.11 | 0.32 | 0.05 | ASI | 5 | 32.73 | 41.89 | 0.02 | 0.03 |
| | 10 | 47.72 | 60.34 | 1.40 | 0.36 | | 10 | 40.51 | 52.12 | 1.36 | 0.63 |
| | | **DLV3 (Chen et al., 2017)** | | | | | | **DLV3-M (Chen et al., 2017)** | | | |
| Vanilla training | All | 58.44 | 72.03 | 1128.02 | 0.97 | Vanilla training | All | 55.87 | 69.47 | 2758.01 | 3.02 |
| | 5 | 40.75 | 52.95 | 336.00 | 0.17 | | 5 | 38.38 | 49.61 | 240.00 | 0.12 |
| | 10 | 55.04 | 69.07 | 560.00 | 0.71 | | 10 | 47.91 | 61.67 | 620.00 | 0.71 |
| Gradient Filter | 5 | 32.18 | 42.93 | 27.47 | 0.17 | Gradient Filter | 5 | 35.70 | 46.71 | 20.71 | 0.12 |
| | 10 | 47.44 | 60.08 | 83.47 | 0.71 | | 10 | 45.40 | 58.97 | 65.62 | 0.71 |
| HOSVD ($\varepsilon = 0.8$) | 5 | 38.52 | 50.14 | 2.66 | 247.63 | HOSVD ($\varepsilon = 0.8$) | 5 | 35.55 | 45.64 | 0.63 | 139.57 |
| | 10 | 50.11 | 63.14 | 1.93 | 392.78 | | 10 | 42.68 | 54.17 | 1.04 | 611.29 |
| ASI | 5 | 38.43 | 50.16 | 2.37 | 0.12 | ASI | 5 | 35.76 | 45.93 | 0.62 | 0.07 |
| | 10 | 45.72 | 58.45 | 1.89 | 0.41 | | 10 | 42.30 | 54.13 | 1.03 | 0.38 |
| | | **FCN (Long et al., 2015)** | | | | | | **UPerNet (Xiao et al., 2018)** | | | |
| Vanilla training | All | 45.36 | 59.53 | 952.00 | 0.90 | Vanilla training | All | 64.71 | 77.32 | 2168.78 | 3.55 |
| | 5 | 27.31 | 38.21 | 288.00 | 0.41 | | 5 | 48.05 | 61.66 | 1380.00 | 3.33 |
| | 10 | 43.54 | 57.96 | 480.00 | 0.72 | | 10 | 48.90 | 63.10 | 1436.00 | 3.35 |
| Gradient Filter | 5 | 27.24 | 38.10 | 18.00 | 0.41 | Gradient Filter | 5 | 46.79 | 60.50 | 33.00 | 3.33 |
| | 10 | 36.91 | 50.14 | 120.00 | 0.72 | | 10 | 47.89 | 62.44 | 36.50 | 3.35 |
| HOSVD ($\varepsilon = 0.8$) | 5 | 26.34 | 36.14 | 1.43 | 206.12 | HOSVD ($\varepsilon = 0.8$) | 5 | 45.28 | 57.80 | 1.35 | 10866.86 |
| | 10 | 30.91 | 41.19 | 3.77 | 295.93 | | 10 | 46.44 | 58.93 | 1.68 | 10881.59 |
| ASI | 5 | 26.51 | 36.26 | 0.99 | 0.23 | ASI | 5 | 42.73 | 54.99 | 1.25 | 1.77 |
| | 10 | 36.44 | 48.05 | 3.65 | 0.43 | | 10 | 45.18 | 58.00 | 1.57 | 1.78 |

Table 4. Performance comparision between vanilla training and ASI when fine-tuning TinyLlama 1B with BoolQ dataset.

| #Layers | Vanilla training | | | ASI (rank=20) | | |
|---|---|---|---|---|---|---|
| | Acc ↑ | Mem (MB) ↓ | TFLOPs ↓ | Acc ↑ | Mem (MB) ↓ | TFLOPs ↓ |
| 1 | 65.94 | 1408 | 3.02 | 64.69 | 0.51 | 1.68 |
| 2 | 66.37 | 1920 | 6.04 | 64.81 | 0.74 | 3.33 |
| 3 | 66.91 | 2432 | 9.07 | 65.00 | 0.98 | 4.98 |
| 4 | 67.44 | 3840 | 12.09 | 66.31 | 1.49 | 6.66 |
| 5 | 67.78 | 4352 | 15.11 | 66.34 | 1.72 | 8.31 |

