# OpenReview forum: "Beyond Low-rank Decomposition: A Shortcut Approach for Efficient On-Device Learning"
_ICML.cc/2025/Conference — ICML 2025 poster_

### Official Review · Reviewer_N3Rd · 2025-03-07

**Overall Recommendation:** 3

**Summary:**

The paper proposes a novel shortcut approach—termed ASI (Activation Subspace Iteration)—that aims to improve the efficiency of on-device learning by addressing the activation memory bottleneck during backpropagation. The key idea is to perform a single subspace iteration with a “warm start” for low-rank decomposition of activation maps, accompanied by a rank selection strategy based on an activation perplexity measure. Experiments on compact networks like MCUNet and other small architectures demonstrate substantial reductions in memory usage and FLOPs compared to vanilla training and traditional HOSVD-based methods.

**Claims And Evidence:**

Yes

**Essential References Not Discussed:**

The paper might benefit from discussing more recent works on scaling low-rank decomposition techniques to transformer architectures and large language models, as these are critical for assessing the broader impact of the proposed method.

**Experimental Designs Or Analyses:**

1. Experiments are conducted on several standard benchmarks and on a Raspberry Pi 5 to validate the on-device feasibility.
2. However, the experiments focus on MCUNet and similar small networks, leaving open the critical question of whether the rank selection and decomposition strategy can scale to transformer-based models or LLMs with billions of parameters.

**Methods And Evaluation Criteria:**

1. The paper’s method centers on a single subspace iteration with a warm start for decomposing activation maps, along with a rank selection mechanism guided by activation perplexity.
2. Evaluation criteria include reductions in activation memory, computational FLOPs, and training latency.
3. A concern is that while the approach shows promise on compact models, the paper does not sufficiently clarify its applicability or scalability when deployed on much larger models such as transformers or large language models (LLMs).

**Other Comments Or Suggestions:**

1. Consider revising the abstract to include more specific methodological details about ASI and the rank selection strategy.
2. Improve the overall writing style and structure to enhance clarity and readability.
3. It is crucial to discuss the limits of the method’s applicability; a dedicated discussion on whether and how the approach could extend to transformer architectures would greatly strengthen the paper.

**Other Strengths And Weaknesses:**

Strengths:
1. The idea of using a single subspace iteration with a warm start for activation compression is interesting and shows promising efficiency gains on small networks.
2. Experimental results demonstrate significant improvements in memory and computational efficiency on resource-constrained devices.

WeaknessesL
1. The abstract and overall writing need improvement; the description of the method is not as clear or detailed as it should be, making the paper hard to read.
2. The applicability and scalability of the approach are not clearly addressed, particularly when considering transformer-based models or large language models. If the rank selection strategy does not scale to LLMs, the impact of the work would be considerably diminished.

**Questions For Authors:**

1. Could you elaborate on how your rank selection strategy might scale to transformer-based models or LLMs with billions of parameters?
2. Have you conducted any preliminary experiments on larger-scale models beyond MCUNet? If so, what were the outcomes; if not, what are your expectations?
3. The paper would benefit from a clearer description in the abstract regarding the methodological details of ASI—could you provide a more detailed outline of the steps involved?
4. Do you foresee any challenges in applying your approach to models with significantly different architectures than those tested in your experiments, and how might you address them?

**Relation To Broader Scientific Literature:**

1. The work builds on prior research in low-rank decomposition for weight and activation compression, including methods like HOSVD and LoRA.
2. It contributes by proposing a shortcut approach (ASI) that potentially reduces computational overhead.
3. However, similar techniques have been explored in previous works, and the novelty may be limited if the approach does not generalize beyond small-scale networks.

**Theoretical Claims:**

1. The authors derive and analyze the computational complexity of ASI compared to HOSVD-based methods, providing equations for memory savings and speedup ratios.
2. Although the theoretical framework is solid, the paper’s writing is dense and could benefit from clearer explanations to enhance reader comprehension.

---

> ### Author Rebuttal · Authors · 2025-03-28
>
> **Question 1: How our rank selection strategy might scale to transformer-based models or LLMs?**
>
> Our rank selection strategy is fully applicable to transformer-based models and LLMs with billions of parameters, and it incurs only a **one-time cost**.
>
> The main idea of our strategy involves performing a brute-force search over a 2D search space, where one dimension represents the number of layers to fine-tune and the other represents the number of explained variance values considered. Increasing or decreasing either of these factors directly impacts the search time for the optimal solution. The total number of parameters or model architecture does not directly affect our rank selection strategy. As long as backpropagation is required for training, it is still applicable.
>
> Furthermore, the choice of rank selection algorithm is not the key point of ASI (*see our response to Weakness 2 of Reviewer dju2*).
>
> **Question 2, Weakness 2 and Suggestion 3: Apply ASI to transformer models and LLMs.**
>
> We have conducted additional experiments as follows:
>
> - **Transformer models**: We applied the same training strategy with ASI implemented on the linear layers within the MLP blocks:
>   - **Swin Transformer for classification tasks**: We used Swin Transformer pretrained on ImageNet-1K from the PyTorch library and evaluated it on five different downstream datasets: CIFAR-10, CIFAR-100, Pets, Flowers, and CUB.
>   - **TinyLlama on BoolQ dataset**: We extended our technique to TinyLlama, a large language model (LLM) with 1.1B parameters, using the BoolQ dataset, which consists of yes/no questions. Since TinyLlama is a very large model, applying $\text{HOSVD}_\varepsilon$ directly would be computationally infeasible given our available resources, making it impossible to construct a budget for comparison. Therefore, instead of using the proposed rank selection strategy, we set a fixed expected rank of **20** for ASI and compared it against vanilla training.
>
> - **Image segmentation**: We applied the same training policy and downstream datasets as used in *(Nguyen et al., 2024)* and *(Yang et al., 2023)*.
>
> These results are available at the following anonymized link: [https://imgur.com/a/qOsfYU5](https://imgur.com/a/qOsfYU5). Overall, ASI gives similar results to those in our submission. It outperforms other methods in both computational complexity and activation memory consumption. We will add these new results to the camera-ready version.
>
> Please note that in addition to MCUNet, we also conducted numerous experiments with MobileNetV2, ResNet18, and ResNet34. The results of these experiments are presented in Tables 1 and 2 of our submission.
>
> **Question 3, Weakness 1, Suggestion 1 and Suggestion 2: Writing style and how ASI works.**
>
> - **Writing style:** Thank you for your feedback. We will carefully consider it and make appropriate revisions.
>
> - **How ASI works:** Regarding the steps in our method, as described in Fig. 1:
>   - **Step 1:** Before the training begins, we measure the perplexity $\mathcal{P}$ (defined in Eq. (7)) of the pretrained model for each predefined explained variance by feeding a minibatch of pretrained data. We then save $\mathcal{P}$ to a file.
>   - **Step 2:** Using the saved file, we run a brute-force search algorithm to find the optimal ranks for each fine-tuned layer. The optimal rank is the one that satisfies Eq. (8) and Eq. (9), i.e., the rank that minimizes the total perplexity while ensuring the memory does not exceed the given budget $\mathcal{B}$.
>   - **Step 3:** We use ASI to "compress" the activation map of each fine-tuned layer into a subspace with the rank corresponding to the one found in Step 2. These ranks remain fixed throughout the training process.
>
>
> **Question 4: Challenges in applying ASI to other architectures.**
>
> We do not foresee any issues applying ASI to models architecturally different than those we have tested. As long as the model requires backpropagation and includes convolutional/linear layers, ASI can still be applied. Would you suggest some architectures where you think ASI might face challenges? We would be happy to hear your thoughts.

---

### Official Review · Reviewer_Ekgz · 2025-03-08

**Overall Recommendation:** 4

**Summary:**

The authors focus on the problem of reducing activation memory usage and computational complexity during on-device learning. The authors try to deploy learning tasks on resource-constrained edge devices while maintaining acceptable performance. The evaluation based on the MCUNet model shows the performance on image classification tasks.

**Claims And Evidence:**

Yes. The claims are easy to follow and the evidence is clearly supported.

**Essential References Not Discussed:**

No. I think the references are adequately covered.

**Ethical Review Concerns:**

N/A.

**Experimental Designs Or Analyses:**

Yes. The experiments are correctly configured and the insights obtained from the experiments are clearly explained.

**Methods And Evaluation Criteria:**

Yes. The authors use typical on-device learning models and image classification tasks.

**Other Comments Or Suggestions:**

Overall, I think this paper is interesting and its technical depth is fine in most aspects.

**Other Strengths And Weaknesses:**

The authors propose a rank selection strategy to determine the most suitable ranks for each fine-tuned layer under a given memory budget constraint before training begins. This is a practical idea to reduce activation memory usage and overall training FLOPs.

**Questions For Authors:**

In experiments, the proposed method reduces overall training FLOPs up to 1.86× compared to vanilla training,. Could you please give more details on how to measure the training FLOPs?

**Relation To Broader Scientific Literature:**

This paper is strongly related to the on-device learning design and its deployment.

**Theoretical Claims:**

Yes. The rank selection and backward pass are clearly formulated.

---

> ### Author Rebuttal · Authors · 2025-03-28
>
> We appreciate your review, below is the answer to your only question.
>
> **Question: How do we calculate training FLOPs?**
>
> Currently, we measure training FLOPs based on theoretical calculations, which consist of the sum of the FLOPs required for both the forward and backward passes. The necessary formulas for a convolutional layer (with similar derivations for linear layers) are derived in equations (13)–(17) of our submission:
>
> - **Vanilla training:**
>   - Forward FLOPs: Defined in Eq. (17).
>   - Backward FLOPs: Defined in Eq. (16).
>
> - **ASI:**
>   - Forward FLOPs: $O_{\text{vanilla}} + O_{\text{ASI}}$, where $O_{\text{ASI}}$ is defined in Eq. (14).
>   - Backward FLOPs: Defined in Eq. (15).
>
> - **$\text{HOSVD}_\varepsilon$:**
>   - Forward FLOPs: $O_{\text{vanilla}} + O_{\text{HOSVD}\varepsilon}$, where $O_{\text{HOSVD}\varepsilon}$ is defined in Eq. (13).
>   - Backward FLOPs: Defined in Eq. (15), which is identical to ASI.
>
> The key reason ASI achieves lower training FLOPs compared to vanilla training is that its low-rank gradient calculation significantly reduces computational costs. This reduction more than compensates for the minor overhead introduced by mapping activation maps to subspaces during the forward pass (see Fig. 5).

---

### Official Review · Reviewer_dju2 · 2025-03-16

**Overall Recommendation:** 3

**Summary:**

This paper proposes Activation Subspace Iteration (ASI), a novel technique to address memory bottlenecks in on-device learning. The method compresses activation maps in neural networks using low-rank decomposition strategies. The key innovations include: (1) a perplexity-based rank selection strategy that identifies optimal compression rates for each layer under a given memory budget constraint before training begins, (2) a single subspace iteration with "warm start" to replace traditional HOSVD-based compression methods, and (3) computation of gradients directly in the compressed space. The authors empirically demonstrate that ASI can reduce activation memory usage up to 120.09× compared to vanilla training while reducing training FLOPs up to 1.86×. On a resource-limited device (Raspberry Pi 5), ASI achieves significant speedups compared to alternative methods (91.0× faster than HOSVD, 1.86× faster than gradient filtering).

**Claims And Evidence:**

The major claims of the paper are generally well-supported by evidence:

1. The activation memory reduction claim (up to 120.09×) is substantiated through extensive experimentation across multiple datasets (CIFAR-10/100, CUB, Flowers, Pets, ImageNet) and architectures (MCUNet, MobileNetV2, ResNet-18/34).

2. The computational efficiency claim (up to 1.86× reduction in FLOPs) is backed by both theoretical analysis (Section 3.5) and empirical measurements (Section 4.3 and 4.4).

3. The accuracy claim (comparable performance to vanilla training and HOSVD) is well-supported through tables and figures showing performance across multiple settings.

**Essential References Not Discussed:**

see Strengths And Weaknesses.

**Experimental Designs Or Analyses:**

see Strengths And Weaknesses.

**Methods And Evaluation Criteria:**

The proposed methods and evaluation criteria are appropriate for the problem of on-device learning:

1. The use of real-world datasets (including ImageNet) is appropriate for evaluating model accuracy.

2. The measurement of memory usage and FLOPs provides clear metrics for resource efficiency.

3. The on-device measurements on Raspberry Pi 5 provide real-world validation of the approach's practicality.

4. The comparison against baselines (HOSVD and gradient filtering) gives context to understand the relative advantages of ASI.

**Other Comments Or Suggestions:**

see Strengths And Weaknesses.

**Other Strengths And Weaknesses:**

Strengths:
1. The paper addresses a critical practical challenge (memory bottleneck) in on-device learning with a novel solution.
2. The paper provides a comprehensive evaluation across multiple datasets, model architectures, and settings.
3. The real-world validation on Raspberry Pi 5 demonstrates that the approach is immediately applicable.

Weaknesses:
1. Authors should use \citet when it is a part of sentence.
2. The rank selection approach relies on a brute-force backtracking algorithm, which the authors acknowledge as a limitation in Appendix C. Is there other possible alternative method?
3. The paper focuses exclusively on convolutional networks and CV task; it's unclear how well ASI would work for other tasks or architectures like transformers.
4. While memory and computation reductions are significant, the accuracy sometimes drops substantially when fine-tuning deeper networks, suggesting scalability limitations.
5. The warm-start strategy assumes that activation maps are stable across training iterations, but this assumption may not hold during early training or when learning rates are high. I am not sure if GaLore can help deal with it?

**Questions For Authors:**

see Strengths And Weaknesses.

**Relation To Broader Scientific Literature:**

see Strengths And Weaknesses.

**Theoretical Claims:**

see Strengths And Weaknesses.

---

> ### Author Rebuttal · Authors · 2025-03-28
>
> **Weakness 1: Use of \citet.**
>
> Thank you for this note. We will revise it in the camera-ready version.
>
> **Weakness 2: Other rank search algorithms besides brute-force?**
>
> Yes, there are certainly alternative methods—for example, using dynamic programming, we might reduce the computational complexity from exponential to linear, at the cost of employing more memory.
>
> However, in the context of on-device learning, the models used are typically small with a limited number of layers, and we are heavily constrained by memory. As a result, brute-force search remains computationally feasible while ensuring an optimal solution.
>
> That said, rank selection itself is not the core contribution of ASI. It would be interesting to explore smarter methods for rank-selection, relying on the learning dynamics of Deep Models (eg. leveraging something like the Information bottleneck); however, this still remains an open research quest. We leave further exploration of rank search algorithms for future work.
>
> **Weakness 3: Apply ASI to transformer models and LLMs.**
>
> We appreciate your feedback. To address this, we expanded our experiments to Swin Transformer, image segmentation, and LLM task using TinyLlama with 1.1B parameters. *For further details, please refer to Question 2 of Reviewer N3Rd*.
>
> **Weakness 4: The problem of accuracy loss.**
>
> We acknowledge that accuracy drops due to compression—this is the tradeoff. As such, this phenomenon also occurs with all other related state-of-the-art techniques, including $\text{HOSVD}_\varepsilon$, Gradient Filter, LoRA, and its variants. The key point is that ASI achieves a superior Pareto curve compared to other methods (Fig. 4).
>
> **Weakness 5: Stability of activation map and combination with GaLore.**
>
> Thank you for suggesting GaLore. Our response is as follows:
>
> - *Stability of activation maps:*
>   - ASI is specifically designed for fine-tuning, where large learning rates are typically not used. As a result, the input to the activation function does not change drastically across training iterations, i.e., it remains stable within a small number of iterations.
>   - Moreover, *(Virmaux & Scaman, 2018)* indicated that most commonly used activation functions have a Lipschitz constant of 1, meaning that if the input remains stable, the output (activation map) remains stable as well.
>
> - *Effect of GaLore:*
>   - Based on our understanding, GaLore does not directly affect the stability of activation maps.
>   - GaLore is a gradient compression technique, while ASI specifically works on activation maps. In principle, both methods could be combined to maximize efficiency.
>   - However, GaLore is particularly useful for Adam, where optimizer state becomes a concern. In contrast, for on-device learning, which is our primary focus, SGD without momentum is preferable since it eliminates the need to store optimizer state. Consequently, GaLore offers little benefit in this setting.
>
> We acknowledge that GaLore is an interesting approach, and leveraging it could lead to promising results-and will be added in the related works section. However, we have not observed how it influences the stability of activation maps in fine-tuning. Could you please clarify your thoughts on this?

---

### Decision · Program_Chairs · 2025-05-01

**Decision:**

Accept (poster)

**Comment:**

This paper presents Activation Subspace Iteration (ASI), a novel technique for addressing memory bottlenecks in on-device learning through efficient compression of activation maps. The work introduces several key innovations, including a perplexity-based rank selection strategy and a single subspace iteration with "warm start" that replaces traditional HOSVD-based compression methods.

The paper has received generally positive reviews, with scores ranging from weak accept to accept. The reviewers consistently praise the paper's practical significance and thorough empirical validation. The method demonstrates impressive results, achieving up to 120.09× reduction in activation memory usage and 1.86× reduction in training FLOPs compared to vanilla training.

During the rebuttal phase, the authors have effectively addressed several key concerns raised by the reviewers. They provided additional experiments demonstrating ASI's applicability to transformer models and LLMs, including results on Swin Transformer and TinyLlama with 1.1B parameters. The authors also clarified their FLOPs calculation methodology and provided detailed insights into the rank selection strategy's scalability.

The authors acknowledge certain limitations of their approach, such as the use of brute-force search for rank selection and potential accuracy drops when fine-tuning deeper networks. However, they have provided reasonable justifications for these design choices, noting that the rank selection is computationally feasible for typical on-device learning scenarios where models are relatively small.

Some writing issues were noted, particularly regarding citation style and abstract clarity. The authors have committed to addressing these in the camera-ready version. The theoretical foundations are sound, though some reviewers suggested the presentation could be more accessible.